# Cost-Effective Method for Using Cross-Species Spike-In RNA for Normalization and Quantification in Polysome Profiling Experiments

**DOI:** 10.3390/genes16111354

**Published:** 2025-11-10

**Authors:** Krishna Bhattarai, Angelo Slade, Martin Holcik

**Affiliations:** Department of Health Science, Carleton University, Ottawa, ON K1S 5B6, Canada; krishnabhattarai@cmail.carleton.ca (K.B.); ziggyslade@cmail.carleton.ca (A.S.)

**Keywords:** spike-in RNA control, polysome profiling, hypertonic stress, Bcl-xL

## Abstract

Background/Objective: Accurate quantification of RNA is critical for RNA-based experiments such as polysome profiling and RT-qPCR. These techniques often rely on control RNA to ensure consistency and reliability across experiments. Commonly used spike-in controls, including in vitro-synthesized mRNA or ERCC mixes, are expensive and time-consuming, limiting accessibility for many laboratories. This study aims to evaluate the use of cross-species total RNA as a cost-effective and reliable spike-in control. Methods: We developed a method using total RNA from a non-homologous species—specifically, yeast RNA—as a spike-in control for experiments involving human cells. The approach was tested across multiple RNA-based assays to assess its impact on quantification accuracy, reproducibility, and interference with endogenous RNA measurements. Additionally, we applied this method to evaluate the translation efficiency of Bcl-xL mRNA in mammalian cells under hypertonic stress. Results: Cross-species spike-in RNA demonstrated minimal interference with experimental outcomes and provided consistent normalization across replicates. The use of yeast RNA enabled accurate fold-change calculations and improved detection of experimental variability. In the case study involving Bcl-xL mRNA, the spike-in control facilitated reliable assessment of translation efficiency under stress conditions. Conclusions: Using total RNA from a non-related species as a spike-in control offers a practical, economical alternative to conventional methods. This approach enhances the reliability of RNA quantification without compromising experimental integrity, making it especially valuable for resource-limited settings, particularly in polysome and RT-qPCR workflows.

## 1. Introduction

RNA molecules are essential to the central dogma of molecular biology, facilitating the transfer of genetic information from DNA to proteins [1]. mRNA translation (or protein synthesis) is a crucial cellular process that is closely tied to cellular functions as well as cell survival [2]. RNA-based experiments help in understanding biological pathways, while RNA-based targets contribute to therapeutic approaches [3,4]. Quantifying mRNA transcripts and assessing mRNA translation efficiency are essential for gene expression studies. Effective methods for gene expression analysis include DNA microarrays, RNA-seq (RNA sequencing) and reverse transcription quantitative polymerase chain reaction (RT-qPCR). Ensuring the fidelity and reproducibility of these experiments necessitates proper controls, as well as maintaining RNA integrity and purity [5,6]. While control RNA serves as a benchmark in RNA-based experiments to ensure reliability and reproducible results, it is important to note that RNA can be lost during isolation from cells and tissues, and different extraction methods have varying efficiencies [7,8,9]. Without proper controls, it is challenging to precisely determine the initial amount of RNA, complicating the quantification of gene expression [10]. Even with meticulous procedures, variations between experiments can occur. Controls provide a reference point, enabling researchers to compare results across different replicates, experiments, and laboratories. Furthermore, controls help identify problems in the entire experimental process, as measured deviations in the spike-in control can identify inconsistencies. The challenge of having inconsistent results between the samples and experiments is amplified in complex experimental protocols such as polysome profiling, which is an example in this paper. Polysome profiling is a technique for assessing mRNA translational efficiency by separating polysomes (a mix of actively translated mRNAs associated with two or more ribosomes) from monosomes (less efficiently or non-translated mRNAs associated with single ribosomes) into discrete fractions [11]. Polysome profiling encompasses various technically non-trivial steps that may introduce experimental variability. These include variations in the efficiency of RNA isolation from each fraction and reverse transcription efficiencies among different fractions. Incorporating spike-in RNA controls aids in normalization, enabling accurate interpretation of RNA expression changes between polysome fractions and samples. Most of the commonly used controls include in vitro-synthesized mRNA from non-related species, such as the chloramphenicol acetyltransferase (CAT) mRNA from bacteria [12] or commercially available External RNA Controls Consortium (ERCC) spike-in mixes as spike-in RNA [13,14]. However, these control RNAs could be prohibitively expensive, posing a significant financial barrier for many research laboratories. Instead, we propose utilizing cross-species total RNA as a control spike-in RNA as a cost-effective method to enhance accuracy in RNA-based experiments. This approach involves employing total RNA from different species (e.g., *Saccharomyces cerevisiae*, tested here), taking advantage of low sequence similarity between species. To demonstrate the feasibility of this approach, we investigated the role of ARK5 in the translation efficiency of the anti-apoptotic gene Bcl-xL in response to hypertonic stress in a human cell line (U2OS). We performed polysome fractionation and used total yeast RNA as a spike-in control to normalize RNA levels in the polysome fractions. Our data shows that the inclusion of cross-species RNA does not interfere with the reverse-transcription of the experiments, while improving data normalization. This approach significantly reduces the cost of experiments while maintaining experimental accuracy and reliability, and demonstrates utility in polysome profiling.

## 2. Materials and Methods


**Minimizing RNA degradation during experimentation**


Ribonucleases (RNases) are ubiquitous enzymes capable of significantly degrading RNA samples, thereby compromising downstream applications. To ensure RNA integrity throughout experiments, strict adherence was maintained to an RNase-free environment, such as consistently wearing and changing disposable gloves throughout the entire experimental workflow, using RNase-free barrier pipette tips and plastics (including tubes, plates, and microcentrifuge tubes), preparing all solutions with nuclease-free water or diethylpyrocarbonate (DEPC)-treated water, and regularly decontaminating surfaces, workspace, and equipment with commercially available RNase inactivation solutions.


**Cell culture and siRNA treatment**


Human osteosarcoma cells (U2OS) were cultured in complete Gibco^TM^ High-Glucose Dulbecco’s Modified Eagle’s Medium (DMEM) (Thermo Fisher Scientific, Ottawa, ON, Canada) supplemented with 10% Heat-Inactivated Fetal Bovine Serum, 1% L-Glutamine, 1 × 10^5^ U/L Penicillin, and 100 g/L Streptomycin at 37 °C with 5% CO_2_.

For the siRNA-mediated knockdowns, 2 × 10^5^ cells were seeded into the wells of a 6-well culture plate and cultured in complete DMEM. Following this, cells were transfected using Lipofectamine^TM^ RNAiMAX transfection reagent (Invitrogen, Burlington, ON, Canada) and the manufacturer’s protocol with 20 nM of siRNAs and maintained for 48 h in DMEM without P/S and then proceeded for further experiments as explained below. The siRNA information is in Table A1.


**Hypertonic stress and cell lysate preparation**


After 48 h of siRNA treatment, hypertonic stress was induced by sorbitol as described previously [15]. Briefly, cells were rinsed with 1X PBS, followed by the subsequent addition of either warmed complete media or 0.6 M D-Sorbitol (Sigma-Aldrich, Oakville, ON, Canada) solution in complete media. The acute stress was performed at 37 °C with 5% CO_2_ for 2 h. After 2 h, to stabilize polysomes, cells were treated with cycloheximide (1 mg/mL) for 10 min, after which they were harvested. For harvesting cells, plates were placed on ice, media was aspirated, and cells were washed with ice-cold PBS containing cycloheximide (100 μg/mL). Subsequently, cells were scraped with ice-cold PBS containing cycloheximide (100 μg/mL) and collected in a centrifuge tube. The collected samples were then centrifuged at 500× *g* for 10 min at 4 °C. The supernatant was discarded, and the pellet was lysed in hypotonic lysis buffer (5 mM Tris-HCl, pH 7.5, 2.5 mM MgCl_2_, 1.5 mM KCl, 1 mg/mL cycloheximide, 2 mM DTT, 0.5% Triton, 0.5% sodium deoxycholate, 200 U/mL RNase inhibitor; all from Sigma Aldrich, Oakville, ON, Canada). The cell lysate was incubated on ice for 10 min with three rounds of vortexing for 5 s each, followed by centrifugation at 12,000× *g* for 10 min at 4 °C. The supernatant (cytosolic extract) was transferred to a new tube and saved for polysome profiling.


**Sucrose gradient preparation and fractionation**


For polysome profiling, a sucrose gradient (10–50%) was prepared in polypropylene tubes (14 × 89 mm) using 50% sucrose at the bottom and 10% at the top. Both sucrose solutions were made in 1× gradient buffer, diluted from the 10X stock gradient buffer (200 mM HEPES, pH 7.6, 1 M KCl, 50 mM MgCl_2_, 1 mg/mL cycloheximide; all from Sigma Aldrich) with water [16]. The sucrose gradients were prepared in Gradient Master^TM^ (Biocomp Instruments, Tatamagouche, NS, Canada). The cytosolic extract was measured using a spectrophotometer (Nanodrop: DeNovix DS-11, Wilmington, DE, USA), and equal amounts based on A260 units (within the range of 10–15 OD, in a maximum volume of 500 μL) were loaded onto the sucrose gradient. Polysome profiles were generated by centrifuging in a SW41 rotor at 39,000 rpm for 2 h at 4 °C in a Beckman Ultracentrifuge (XPN-100, Mississauga, ON, Canada). Gradients were fractionated (20 fractions, collected 500 μL per tube), and absorbance at 254 nm was recorded.


**Preparation of spike-in RNA**


5 mL of mid-exponentially growing yeast *S. cerevisiae* cells (OD600, 0.3–0.6) were centrifuged in a 15 mL conical tube at 3000× *g* (room temperature) for 3 min. The supernatant was decanted, and the cells were frozen in liquid nitrogen. To the frozen cell pellet, 1 mL of Trizol (Thermo Fisher Scientific, Ottawa, ON, Canada) and 200 μL of 0.5 mm acid-washed disruption beads were added. The solution was then vortexed for 40 s and placed on ice for 1 min for 5 rounds. 1 mL of the lysate was then transferred to a 2 mL tube, 200 μL of chloroform was added, vortexed, and left to rest at room temperature for 5 min. The sample was then centrifuged for 15 min at 12,000× *g* in a microcentrifuge at 4 °C. The top aqueous phase was transferred into a 2 mL tube containing 800 µL of isopropanol and vortexed. The sample was then left to precipitate the RNA overnight at −20 °C. RNA was pelleted at 12,000× *g* for 15 min in a microcentrifuge at 4 °C. The supernatant was carefully decanted, and 1 mL of 70% ethanol was added to wash the pellet. The mixture was centrifuged at 12,000× *g* for 5 min at 4 °C. Ethanol was carefully decanted. The samples were then left to air dry in an RNase-free culture hood for 30 min. The pellet was resuspended in 50 µL RNase-free water by pipetting up and down. RNA concentration was measured using a spectrophotometer (Nanodrop, DeNovix DS-11, Wilmington, DE, USA). The A260/A280 ratio of purified RNA should be greater than 2.0, and the A260/A230 ratio should be approximately 2.2 or slightly higher, with a clear peak around A260. The RNA was stored at −80 °C.


**Addition of Spike-in RNA, RNA isolation, cDNA preparation, and RT-qPCR**


After polysome fractionation, a total of 320 ng of yeast total RNA (spike-in RNA) was added to each fraction before RNA isolation. Total RNAs were isolated from the fractions by adding 25 µL of Proteinase K solution for every 500 µL of sucrose fraction (18.75 µL 10% SDS, 3.75 µL 0.5 M EDTA, 0.5 µL Glycoblue, 2 µL of 20 mg/mL Proteinase K) and incubated at 55 °C for 1 h. Then, an equal volume of phenol/chloroform/isoamyl alcohol (125:24:1, preferably acidic, pH 4.5) was added to the sucrose fractions. An extra 100 µL of chloroform was added to fractions 14–20, as these heavier, high-sucrose fractions are more viscous and prone to phase inversion. After vortexing for 30 s, the mixture was centrifuged at maximum speed for 5 min at room temperature. Then, 80–90% of the aqueous phase was saved in a new tube. An equal volume of chloroform was added, and the vortex/centrifuge steps were repeated. Again, the aqueous phase was saved in a new tube and 1:10 volume of 3 M sodium acetate (pH 5.2) and 1.5 volumes of chilled, absolute ethanol were added, followed by vortexing for 15 s. The mixture was precipitated overnight at −20 °C, followed by centrifugation at maximum speed for 30 min at 4 °C the next day. The pellet was washed with 1 mL of chilled RNase-free 70% ethanol, then air dried, and resuspended in 20 µL of RNase-free water. The isolated RNA was then used for cDNA synthesis.

The associated mRNAs were transcribed to cDNAs using the SuperScript™ IV cDNA synthesis kit (Invitrogen, Burlington, ON, Canada). Quantitative PCR (qPCR) was performed using gene-specific primers, SsoAdvanced^TM^ Universal SYBR Green Supermix (Bio-Rad, Mississauga, ON, CA), and CFX96^TM^ Real-Time System (Bio-Rad, Mississauga, ON, CA). The primer information is in Table A1. The mRNAs were normalized to the RLP24 gene of *S. cerevisiae* total RNA used as spike-in RNA. The quantification cycle (Cq) of mRNAs was normalized relative quantity analyzed with CFX Maestro 1.1 version: 4.1.2433.1219 software (Bio-Rad Mississauga, ON, CA).


**Polysome profiling data analysis**


Polysome profiling data were analyzed using normalized relative quantity to calculate the percentage of RNA signal across polysome fractions for the indicated mRNAs. The percentage was used to determine the weighted average (F_W_) for each mRNA distribution. The difference in weighted average (ΔF_W_), representing the shift in mRNA distribution across polysome fractions, was calculated by subtracting the F_W_ of untreated (DMEM) samples from that of the 0.6 M sorbitol-treated (SOR) samples, as described in [17]. Briefly, F_W_ was calculated as: $F_{W}=\frac{\Sigma i\left(fi\times pi\right)}{100},$ where *fi* is the fraction number and *pi* is the RNA signal expressed as a percentage of total mRNA in fraction *i*. RNA signal can be calculated either with spike-in normalization (percentage of gene expression in each fraction normalized to spike-in yeast RPL24) or without normalization (percentage of gene expression per fraction). ΔF_W_ was calculated as: ΔF_W_ = F_W_ (SOR) − F_W_ (DMEM). Fractions 2–16, covering 40S, 60S, 80S, and polysomes, were used in the calculation. The graph was generated using GraphPad Prism 10.4.2 software.


**Statistical analysis**


Data were analyzed using a one-tailed Student’s *t*-test with the GraphPad Prism 10.4.2 software and are expressed as a mean ± standard error of the mean of at least three independent experiments.

## 3. Results

### 3.1. Cross-Species Spike-In RNA Does Not Interfere with Target mRNA Quantification

To verify the specificity of species-specific primers and to confirm that the addition of cross-species RNA does not impair amplification of target mRNAs, we used primers targeting human genes (FN1, Bcl-xL, GAPDH, RPL13A) on yeast cDNA samples and human cDNA samples supplemented with spike-in yeast cDNA for PCR amplification. As expected, PCR amplification was observed only in the human cDNA sample supplemented with spike-in yeast cDNA, as shown in Figure 1a. Similarly, when we used primers targeting yeast genes (CCW12, HHT2, ISO2, MRPL39, RDS3, RLP24, RNP1, RPP2A) on a human cDNA sample or human cDNA sample supplemented with spike-in yeast cDNA, PCR amplification was observed only in the human cDNA sample supplemented with spike-in yeast cDNA, as shown in Figure 1b. Moreover, when PCR amplification on human cDNA alone and the human cDNA sample supplemented with spike-in yeast cDNA were compared, no significant reduction in the efficiency of amplification was observed in the human cDNA sample supplemented with spike-in yeast cDNA, as shown in Figure 1c. However, a slight variability in the quantification of RPL13A was observed between the human cDNA alone and the human cDNA sample supplemented with spike-in yeast cDNA, which may reflect minor fluctuations in amplification efficiency when mixed cDNA templates are used.

This result suggests that the addition of yeast cDNA, even in significant excess, does not negatively impact the amplification of human target mRNAs.

### 3.2. Spike-In RNA Reveals Inconsistencies in RNA Isolation from Polysome Fractions

To identify RNA loss during the isolation process and apply this information for normalization of target gene expression, we utilized a cost-effective cross-species RNA. Specifically, total RNA from yeast (*S. cerevisiae*) was used as a spike-in control during polysome profiling of human cells. Figure 2 illustrates the amplification and relative quantity of yeast RLP24 mRNA, in polysome profiling fractions of U2OS cells that were spiked with total yeast RNA. Interestingly, despite the addition of an equal amount of yeast total RNA (spike-in RNA) to each fraction before RNA isolation, Figure 2 demonstrates inconsistencies in RNA isolation between fractions. This observation underscores the importance of using spike-in RNA normalization control during RNA isolation processes.

### 3.3. Hypertonic Stress Increases the Translation Efficiency of Bcl-xL in siARK5 Treated Cells

To validate the concept of using cross-species spike-in RNA in polysome profiling, we set out to assess the translation efficiency of Bcl-xL mRNA during hypertonic stress. We have shown previously that Bcl-xL protein expression increases during hypertonic stress and is dependent on the activity of ARK5 kinase [15]. However, whether this increase was due to increased translation of Bcl-xL mRNA was not determined. We therefore evaluated the mRNA translation efficiency of the Bcl-xL mRNA using polysome profiling. The initial step involved silencing of ARK5 using siRNA, followed by subjecting cells to hypertonic stress or leaving them untreated. Polysome profiling and data analysis were performed as described in the Section 2.

As shown in the polysome profile (A254) traces (Figure 3a), global translation was reduced under hypertonic stress induced by sorbitol treatment. However, silencing of ARK5 shifted the distribution of Bcl-xL mRNA toward heavier polysomes under stress (Figure 3c,d). This effect was further quantified by calculating the weighted average (ΔF_W_), which confirmed enhanced translational efficiency of Bcl-xL in response to sorbitol treatment (Figure 3e). This finding aligns with previous research demonstrating that silencing ARK5 enhances Bcl-xL expression in response to hypertonic stress [15]. However, when we compared the ΔF_W_ values with and without spike-in normalization, as shown in Figure 3e, the spike-in normalization resulted in a more statistically discernible outcome (lower *p*-value). In addition, to evaluate the effectiveness of paired comparisons between ΔF_W_ values of siRNA treatments with and without spike-in normalization, the correlation coefficient (r) for the spike-in condition was 1.00 with a one-tailed *p*-value of 0.0015, indicating a highly effective pairing. In contrast, the correlation coefficient for the condition without spike-in was 0.9389 with a one-tailed *p*-value of 0.1118, suggesting a less effective pairing.

## 4. Discussion

We have demonstrated that a cross-species spike-in control RNA provides an affordable alternative without compromising accuracy or reliability. RNA from various species could be used as cross-species RNA controls due to their distinct genetic makeup and ease of accessibility. When the species used for control RNA is evolutionarily distant from the target species, the RNA sequences are less likely to share homologous regions [18]. Yeast RNA can be used as a spike-in control in experiments involving mammals such as humans and mice [19]. On average, yeast and their human homologs exhibit only 31% identity in protein-encoding genes [20]. Therefore, yeast RNA sequences are sufficiently different from those of higher organisms, minimizing the risk of cross-reactivity. Similarly, RNA from invertebrates such as *Caenorhabditis elegans* can be employed as a control in experiments involving vertebrate cells. Its well-characterized genome and evolutionary distance from vertebrates make it a suitable choice for cross-species controls. On average, *C. elegans* and its human orthologs exhibit only 49.1% identity at the protein level [21]. Additionally, RNA from the fruit fly (*Drosophila melanogaster*) is distinct from mammalian RNA, making it a versatile control for mammalian cells and tissue experimental setups. The *Drosophila* genome shares 60% homology with the human genome [22].

One of the primary concerns in RT-qPCR is the specificity of the primers used for amplification. If the primers are not specific enough, they could anneal to both human and yeast RNA sequences, leading to non-specific amplification and erroneous quantification. To address this concern, we used primers that are specific to the target human genes. The results, as shown in Figure 1a, confirmed that the primers exclusively amplified human cDNA without cross-reacting with yeast RNA. This ensures that the quantification of human mRNAs is accurate and not influenced by the presence of yeast RNA. Another primary concern is whether yeast spike-in RNA has the potential to interfere with the amplification of human mRNAs. This caveat arises from the possibility that the yeast RNA could compete with human RNA during the reverse transcription and subsequent amplification steps, potentially leading to inaccurate quantification of human transcripts. Studies have shown that cross-species RNA does not interfere with the amplification and quantification of target RNA sequences in the host species [18,19]. Our study also demonstrates that using yeast RNA as a spike-in control in human cell experiments does not negatively impact the amplification of human mRNAs, as shown in Figure 1b. This ensures that the presence of control RNA does not distort the experimental outcomes, maintaining the integrity of the data.

Cross-species RNA can be used across various RNA-based experiments, particularly polysome profiling and RT-qPCR [18,19,23,24]. Its versatility makes it a valuable tool in different research contexts, allowing researchers to apply consistent controls across multiple experimental setups. By including cross-species RNA as a control, researchers can better identify and account for technical variability in their experiments. For example, inconsistencies in RNA isolation, as shown in Figure 2, can be detected and corrected, leading to more accurate quantification of gene expression [25,26,27]. This helps in troubleshooting and refining experimental protocols. With cross-species RNA controls, results can be more easily compared across different experiments and laboratories. This standardization is essential for validating findings and ensuring that research outcomes are reproducible and reliable, contributing to the overall robustness of scientific research. For instance, in our previous study [15], ARK5 knockdown (~80%) increased Bcl-xL protein by 78%, correlating with a ΔF_W_ shift and confirming enhanced translation of Bcl-xL (Figure 3c–e), demonstrating how cross-species RNA controls improve the accuracy and reliability of translational measurements.

The cost-effective nature of this method compared to synthetic RNA or commercially available spike-in controls makes RNA profiling accessible to a broader range of laboratories and enables broader access to high-quality experimental controls, facilitating more extensive and frequent experimentation. Cross-species RNA spike-ins effectively control for variations and errors commonly encountered in RNA-based experiments. The method is straightforward to implement using existing laboratory techniques and equipment, requiring no significant additional training or investment. This method offers a straightforward and generally less labour-intensive approach compared to the in vitro transcription approach, increasing laboratory efficiency. Researchers can seamlessly integrate this method into their current workflows. These controls aid in standardizing data across diverse samples and experimental conditions, thereby mitigating variations in RNA isolation and reverse transcription efficiencies. Overall, the adoption of cross-species RNA as control RNA in experiments addresses both economic and scientific challenges, fostering high-quality, reproducible research that can advance our understanding of biological processes and disease mechanisms.

## Figures and Tables

**Figure 1 genes-16-01354-f001:**
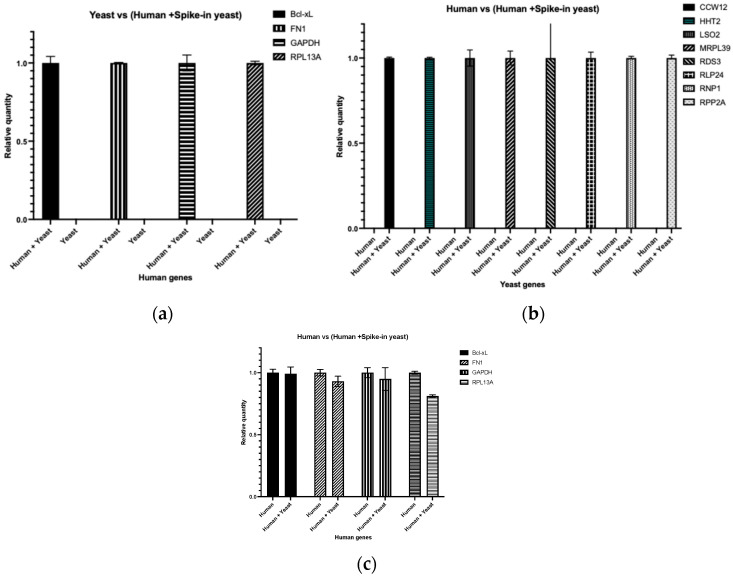
Yeast spike-in RNA maintains specificity and does not significantly interfere with the quantification of human RNA. (**a**) Relative quantity of indicated human genes using a yeast (*S. cerevisiae*) cDNA sample and a human (U2OS cells) cDNA sample supplemented with spike-in yeast cDNA; (**b**) Relative quantity of indicated yeast genes using a human (U2OS cells) cDNA sample and a human (U2OS cells) cDNA sample supplemented with spike-in yeast cDNA; (**c**) Relative quantity of indicated human genes using a human cDNA sample and human cDNA sample supplemented with spike-in yeast cDNA. The relative quantity (ΔCq) of the target gene in each sample, relative to the amount of cDNA template loaded, was calculated using Bio-Rad CFX Maestro software (Maestro 1.1 (4.1.2433.1219)).

**Figure 2 genes-16-01354-f002:**
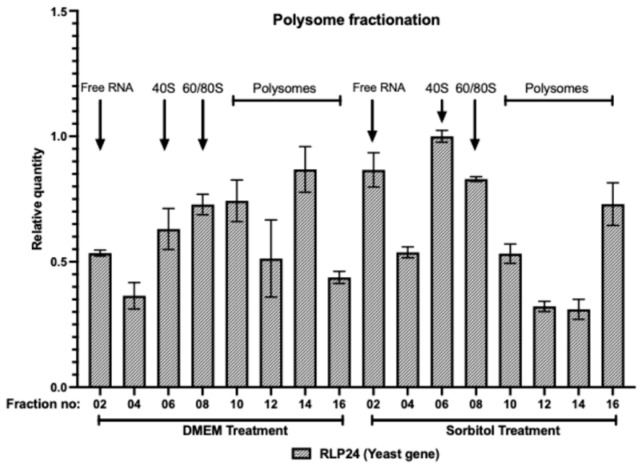
Polysome fractions show inconsistencies in RNA isolation. U2OS cells were treated with DMEM (No treatment) or 0.6 M sorbitol (SOR) for 2 h. Cell lysates were collected and centrifuged in a sucrose gradient (10–50%). Gradients were fractionated (500 µL per tube), and absorbance at 254 nm was recorded. A total of 320 ng of *S. cerevisiae* total RNA (spike-in RNA) was added to each fraction before RNA isolation. RNA was isolated, and the relative quantity of the yeast gene (RLP24) from fractions was determined. The relative quantity (ΔCq) of the target gene in a sample, relative to the amount of cDNA template loaded, was calculated using Bio-Rad CFX Maestro software. The *X*-axis represents the fraction number and treatment.

**Figure 3 genes-16-01354-f003:**
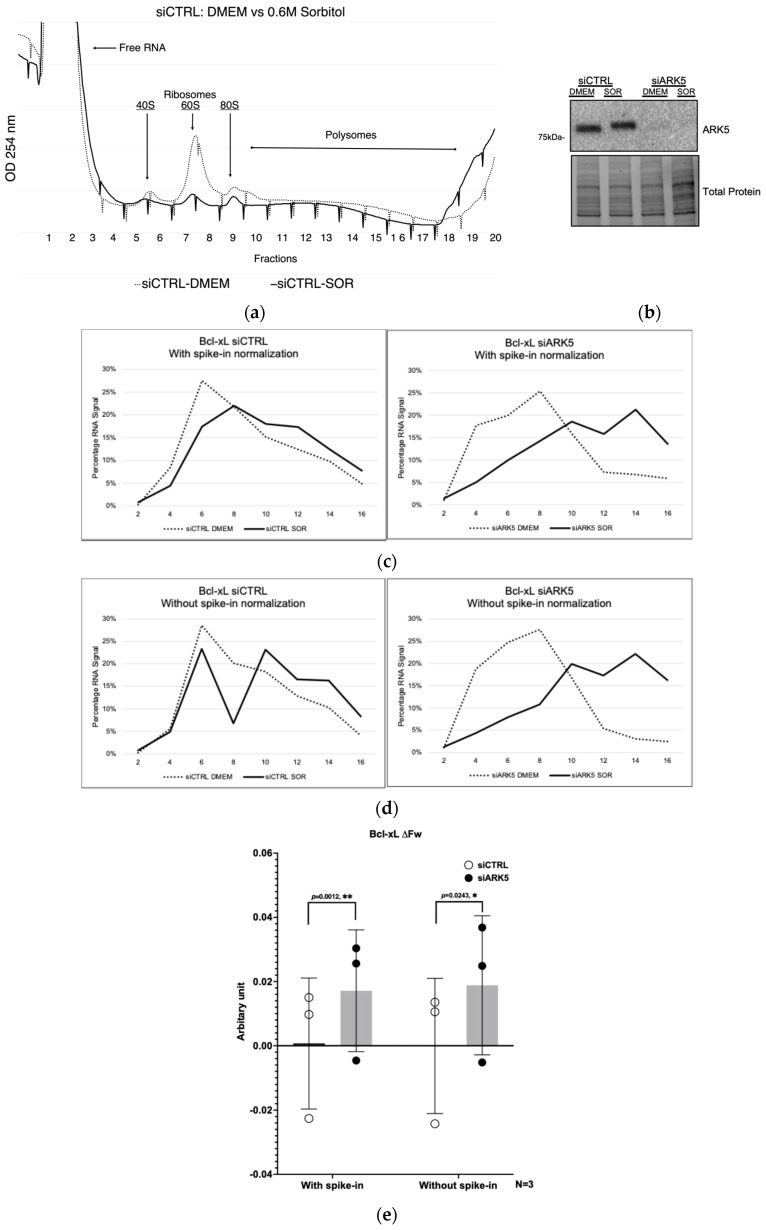
ARK5 impacts the Bcl-xL mRNA translation efficiency in hypertonic-stressed U2OS cells. (**a**) U2OS cells were transfected using siCTRL or siARK5 for 48 h and subsequently treated with DMEM (No treatment) or 0.6 M sorbitol for 2 h. The *X*-axis represents the fraction number. Polysome profiles were analyzed by sucrose gradient (10–50%) centrifugation. Gradients were fractionated (20 fractions, collected 500 µL per tube), and absorbance at 254 nm was recorded. A total of 320 ng of *S. cerevisiae* total RNA (spike-in RNA) was added to each fraction before the RNA isolation. (**b**) Representative immunoblots of cell extracts probed with ARK5 antibody to verify silencing of ARK5. Stain-free gels were used as a loading control for total protein. (**c**,**d**) Representative percentage RNA signal of fractions 2–16 (even fractions only), where Bcl-xL mRNA levels were determined by RT-qPCR in equal volumes of fractions from A. Normalized relative quantity (ΔΔCq) of the target gene relative to the RLP24 (**c**) and relative quantity (ΔCq) of the target gene (**d**) was calculated using Bio-Rad CFX Maestro software. (**e**) The difference in translation efficiency (calculated as weighted average, ΔF_W_) was determined from repeated experiments depicted in (**c**,**d**). The ΔF_W_ for the distribution of the Bcl-xL mRNA level was determined for each percentage of RNA signal associated with the Bcl-xL, following the calculation described in the Section 2 (* *p* < 0.01, ** *p* < 0.05; data were analyzed using one-tailed paired *t*-test).

## Data Availability

The original contributions presented in this study are included in the article. Further inquiries can be directed to the corresponding author.

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
