# Peer review of "Cost-Effective Method for Using Cross-Species Spike-In RNA for Normalization and Quantification in Polysome Profiling Experiments"

_genes, 2025, doi:10.3390/genes16111354_

Round 1
Reviewer 1 Report
Comments and Suggestions for Authors
In the manuscript entitled “Cost-effective method for using cross-species spike-in RNA for normalization and quantification in polysome profiling experiments” by Bhattarai et al., proposes a low-cost normalization strategy for polysome profiling using cross-species total RNA (yeast) added before RNA isolation. They show (i) primer sets maintain species specificity using cDNA spike-ins (Figure 1); (ii) equal yeast spike-in across fractions reveals extraction variability (Figure 2); and (iii) application to Bcl-xL translation under hypertonic stress with/without ARK5 knockdown, where spike-in normalization improves statistical discernment of ΔFW (Figure 3E). Overall, the idea is practical and addresses a critical point for translational profiling labs. Methodological transparency and comparative benchmarking need to be strengthened to support broad adoption.
Major comments
- Table S1 lists multiple yeast genes (e.g., CCW12, HHT2, LSO2, MRPL39, RDS3, RLP24, RNP1, RPP2A) used as potential spike-ins/assay targets. Please justify how the normalizer(s) were chosen (abundance, stability, amplicon efficiency), and compare at least two or three yeast spike-in genes side-by-side as normalizers to quantify between-spike variability (report CVs).
- Since the central claim is “cost-effective alternative” to ERCC or in-vitro transcripts, include a head-to-head comparison on the same gradients (variance, ΔFW, effect sizes, p-values) against one commonly used control. This will substantiate the economic and analytical benefit asserted in Abstract/Discussion.
- Figure 1 uses cDNA spike-ins to show no interference with human amplicons, which does not assess potential reverse-transcription-level competition. Please add RNA-level titrations (human + yeast RNA mixed before RT) to test whether yeast RNA affects human cDNA yield/Cq.
- Methods state ΔFW is “as described in [16]” but do not show the formula or exact fraction range used in the calculation. Please provide the equations for FW and ΔFW, define inputs, and justify ΔFW vs alternative metrics (e.g., P/M ratio).
- Add rRNA (28S/18S) gels or Bioanalyzer traces (or annotated A254 with 40S/60S/80S boundaries) to document resolution and to contextualize the selected “even fractions 2–16” used for qPCR quantification.
- There’s a discrepancy in fraction volumes: Figure 2 legend states 1 mL/tube, while Figure 3 and Methods specify 20 fractions at 500 µL/tube. Please reconcile and standardize the reported volumes.
- Methods specify adding an extra 100 µL chloroform to fractions 14–20. Briefly justify (e.g., viscosity/high sucrose) and confirm that extraction efficiency is uniform across the gradient (control for recovery).
- You add 320 ng yeast RNA per fraction before isolation. Please justify this dose with titration/linearity data showing no saturation and minimal noise inflation, and indicate whether the same dose is appropriate for other cell types/RNA yields.
- You use one-tailed paired t-tests. Provide the a priori directional hypothesis or switch to two-tailed tests; report n for each analysis and exact p-values.
- In Figure 3C–D, shapes look broadly similar with/without normalization, yet ΔFW significance improves with spike-in. Expand the explanation (variance reduction, pairing effectiveness). Consider showing variance components or Bland-Altman plots.
- Provide in-silico alignments (e.g., BLAST-like) of each yeast primer pair against the human transcriptome (top hits, %ID, E-values) and include melt curves/gels confirming single products for all amplicons used. (Figure 1 and Table S1 list the sets.)
- Data are currently “available upon request.” Please deposit raw Cq tables, primer sequences (already in S1), full A254 traces, and analysis scripts for FW/ΔFW in a public repository to meet reproducibility standards.
Minor comments
- Figure order and cross-references: Ensure figures are introduced in numerical order and correct any mismatches (e.g., places where Figure 2A/2B are cited for content that appears in Figure 1; where Figure 3E is referenced before 3A–D).
- Axis labeling/coverage: In Figure 3A, add fraction numbers on the x-axis to match panels C–D; confirm that all indicated even fractions 2–16 are consistently shown across panels (address any missing “fraction 2” instances).
- Terminology: Replace “discreet fractions” with “discrete fractions.”
- Methods clarity (buffers): The “1× gradient buffer” lists unusually high final concentrations (200 mM HEPES, 1 M KCl, 50 mM MgClâ‚‚). Confirm whether these are stock values or typographical, and report the actual finalbuffer composition used during gradient formation.
- Application example details: State the % ARK5 knockdown (mean ± SEM) and whether Bcl-xL changes at protein level correlate with the observed ΔFW shift.
- Figure 2 context: Include the corresponding A254 traces for the exact gradients used to generate the RLP24 measurements.
- Software/versioning: CFX Maestro and GraphPad Prism are mentioned, specify exact versions already used (some places do; make this consistent across sections).
Reviewer 2 Report
Comments and Suggestions for Authors
This paper could be a part of a bigger manuscript dedicated to some sound and scientifically relevant study. But it is not big enough to be published on its own as a method. if falls short in a number of measurements, statistical analysis, all possible controls and extrapolations to other circumstances. It is simply too small to stand on its own.
In this particular case I do not see a need to go into any possible small details of this manuscript.
Reviewer 3 Report
Comments and Suggestions for Authors
The authors present, with some justification, the proposal to use yeast RNA as a spike-in for mammalian studies to allow for normalization between experiments and possible quantitation. The data in Figure 1 are quite convincing that there is no cross contamination in either direction (yeast to human and visa versa) and the yield of cDNA is essentially the same for several proteins although the L13a is down 20% in panel c. Thus, one of the elements not addressed is just exactly how good is the quantitation for the yeast spike-in (i.e. +/- 5%, 10%, etc.)? A larger concern arises in Figure 2 where the relative recovery of the yeast spike-in seems to vary by 3-fold.
Concerns
- While the use of the yeast RNA seems OK experimentally, it is not clear why coli RNA is not used which would be expected to be even less similar to mammalian RNA and thus might yield cleaner signals.
- The loss of the yeast spike-in in Figure 2 means large corrections would be required (up to 3-fold). Is there any reason why the authors have not tried to use larger amounts of RNA to facilitate more quantitative recovery of both the spike-in RNA as well as the mammalian RNA under study? Also, the authors have not indicated how much mammalian RNA is present to begin their experiments (as A260 or ng)
- For the quantitative aspect, should there be an original comparison of total RNA to DNA as the DNA would represent one copy per cell (depending where in the cell cycle the cells are harvested)?
- Figure 3 – panel a does not appear to indicate any polysomes compared to what normal polysome profiles look like. Is there a reason for this? Secondly, it would help the reviewer to know where 80S ribosomes and polysomes are located in this gradient. Additionally, the type size around the figures should be increased to facilitate reading.
Minor Concerns
- The authors need to be consistent in their use of abbreviations. Use either mL or ml but not both. For time units, use either full names or standard abbreviations (i.e. use either seconds or sec).
- The sucrose gradient solutions seem to have elevated concentrations of both Mg+2 (50 mM) as well as KCl (1 M). Most others use half (or less) of these two components in their gradients. Is there a reason for these higher concentrations?
Round 2
Reviewer 1 Report
Comments and Suggestions for Authors
The manuscript is substantially improved, and I thank the authors for the efforts they put into responding to the reviews. I remain concerned, however, about the lack of direct comparisons between synthetic and yeast RNA as efficient spike-ins. While yeast RNA is useful for PCR, it is not practical for RNA-seq, which is increasingly used as the go-to method for global translation profiling.
Reviewer 2 Report
Comments and Suggestions for Authors
It is still the same set of data. If you insist you can publish with all consequences attached.
Reviewer 3 Report
Comments and Suggestions for Authors
Reviewer 3 – response to comments
The rational for using yeast RNA rather than E. coli RNA is not well explained. It would seem that ideally one would use an RNA least likely to have sequence identity. Secondly, the variable recovery of RNA in Figure 2 would suggest a possible three-fold correction. Could this be solved by the combination of both a spike-in RNA as well as carrier RNA (or even carrier DNA)? Third, the polysome profile just doesn’t make sense (Figure 3a). There does not appear to be much in the way of polysomes in the control (and the use of 10 to 15 A260 per tube they should be easy to find). Secondly, the sorbitol stress would be anticipated to trigger a breakdown of polysomes which should lead to a larger 80S peak than what is seen in the control. This is not the case here – why?
Other concerns
- Figure 1c – the authors should note in the text there is some variability in quantification (RPL13A).
- In Figure 2, is the recovery of material related to where the greatest amount of A260 is in the gradient (there are no gradient markers in the figure)?
- Editorial corrections – time units – see lines 133,194, and 200. When expressing concentrations, volumes or temperature put a space between the number and the unit (i.e. 1 ml, 4o C, 3 mM). See lines 169, 170, 173, 183, 200 and possibly others. Figure 2 legend – “Gradients were fractionated (500 ml per tube)…”
Round 3
Reviewer 3 Report
Comments and Suggestions for Authors
The authors have addressed most of this reviewer's concerns except for the polysome profiles in Figure 3a.